# Trends and factors associated with pneumonia mortality at Buea Regional Hospital in Cameroon from 2020 to 2023

Kukwah Anthony Tufon[1,2]*, Tiayah Patience Foumene[1,2], Malika Esembeson[2,3], Ayah Flora Bolimo[1,2], Tiayah Precious Yemene[4], Chapajong Mariecole Nguatem[2], Teuwafeu Denis Georges[2,5], Ronald Mbua Gobina[2,5], Nkouonlack Cyrille[2,5], Tiayah Munge[2], Ngomba Divine Martin Mokake[2,6]

**1** Department of Microbiology and Parasitology, Faculty of Science, University of Buea, Buea, Cameroon, **2** Buea Regional Hospital Buea, Buea, Cameroon, **3** Department of Public health and Hygiene, Faculty of Health Sciences, University of Buea, Buea, Cameroon, **4** Department of Pharmacy, Faculty of medicine and pharmaceutical science, University of Dschang, Dschang, Cameroon, **5** Department of Internal Medicine, Faculty of Health Science, University of Buea, Buea, Cameroon, **6** Department of Surgery and Specialties, Faculty of Health Sciences, University of Buea, Buea, Cameroon

* drkapt@yahoo.com

## Abstract

### Background

Pneumonia cases and associated deaths remain a public health concern necessitating ongoing efforts in prevention, diagnosis, and treatment. This study was aimed at identifying the trend of pneumonia, case fatality rate and factors associated with pneumonia mortality at the Buea regional hospital from 2020 to 2023.

### Methods

In this retrospective study, data was collected from all available and complete adult inpatient/outpatient hospital records from 2020 to 2023 at Buea regional hospital. Only patients clinically and radiologically diagnosed with pneumonia (excluding TB and COVID-19) were considered for this study. Data was analyzed using SPSS with statistical significance set at $p < 0.05$.

### Results

After reviewing 18,805 entries/records, 746 files corresponding to 746 patients (59.5% females) diagnosed with pneumonia were retained with mean age of 54.7 ± 19.2 years. There was a fluctuating trend in cases across the years with 2021 generally having higher values. There was a decreasing trend in case fatality rate across the years (2020: 17%, 2021:24%, 2022:8%, 2023:6%) with an overall case fatality rate of 13.7%. Being hospitalized for ≤10 days (aOR: 10.14, CI: 2.31–44.52, p = 0.002) and not being treated with amoxicillin-clavulanic acid (aOR: 3.33, CI: 1.43–7.75, p = 0.005) were significantly associated with pneumonia mortality.

**Data availability statement:** All relevant data are within the manuscript and its Supporting information files.

**Funding:** The author(s) received no specific funding for this work.

**Competing interests:** The authors have declared that no competing interests exist.

## Conclusion

Although the number of pneumonia cases fluctuated over time, they remained consistently present across all months from 2020 to 2023, highlighting the year-round nature of pneumonia transmission and its endemicity in the Buea Health District. Being hospitalized for less than 10 days and not using amoxicillin-clavulanic acid for treatment were risk factors of pneumonia mortality. These findings underscore the importance of timely diagnosis and empiric treatment with guideline recommended agents such as amoxicillin–clavulanic acid so as to reduce pneumonia mortality in the Buea regional hospital.

## Introduction

Pneumonia affects people of all age groups and is a major public health problem worldwide. It contributes significantly to high mortality and morbidity rates, especially in children under five years of age and in older adults [1]. Pneumonia has been reported to have an estimated global incidence ranging between 1.5 to 14 cases per 1000 person-years [2]. Research shows that 95% of the total number of pneumonia cases occur in low-and middle- income countries (LMIC) [3]. In Sub-Saharan Africa, 4 million cases of pneumonia occur each year resulting in 200,000 deaths [4]. Risk factors in adults include: age > 65 years, smoking, alcoholism, immunosuppressive conditions, chronic obstructive pulmonary disease, cardiovascular disease, cerebro-vascular disease, chronic liver or renal disease, diabetes mellitus and dementia [5].

Pneumonia is an infection characterized by a sudden inflammation of the lung air sacs (alveoli) and other structures of the lung parenchyma., which causes patients to have fever, cough, shortness of breath and malaise [6]. It can be caused by several bacteria, viruses and fungi [7]. Specific examples of causative agents include *Streptococcus pneumoniae, Staphylococcus aureus, Haemophilus influenzae* [8], *Legionella pneumophila, Mycoplasma pneumoniae, Chlamydophila pneumoniae,* and viruses [9].

This study was aimed at identifying the trend of pneumonia and factors associated with pneumonia mortality at the Buea regional hospital from 2020 to 2023. It is hard to find reliable and consistent data on pneumonia over a prolonged period of time in many countries including Cameroon [6]. Generating information on the trend of pneumonia can help in understanding how cases fluctuate over a given period of time and hence help in predicting future outbreaks. This information can add to the literature of pneumonia in Cameroon and also help in planning and allocating resources for the control of pneumonia in Buea Health District (BHD).

Pneumonia cases and associated deaths remain a public health concern necessitating ongoing efforts to improve on the prevention, diagnosis, and treatment [10]. Determining the case fatality rate and identifying factors associated with mortality from pneumonia can help provide information on how severe the disease condition is and guide treatment protocols aimed at optimizing the chances of survival of pneumonia patients. Identifying risk factors associated with mortality can also provide

insights into the underlying causes of mortality, enhance surveillance efforts by focusing on high-risk groups (prioritize early detection and response) and guide scientific investigations and innovations. This study aimed to determine the trends and factors associated with pneumonia and its case fatality rate at Buea Regional Hospital from 2020 to 2023.

## Materials and methods

### Study site, design and population

This retrospective study was carried out at the Buea regional hospital (BRH) within the Buea health district (BHD) situated in Fako division of the Southwest region of Cameroon. Buea is the administrative capital of the Southwest region of Cameroon. Buea regional hospital was purposively selected for this study because it is the largest most equipped hospital and has the highest number of medical doctors in BHD including a pulmonologist, making it ideal for diagnosing and managing pneumonia. This hospital receives many patients on a daily basis as compared to other hospitals or clinics in BHD. Data was collected from all available and complete adult inpatient/outpatient hospital records from 2020 to 2023 at BRH. The inclusion and exclusion criteria were as follows:

**Inclusion criteria.**

i.  Adult patients (≥18 years) who attended the Buea regional hospital (outpatient or inpatient) between January 2020 and December 2023.

ii.  Patients with a clinical diagnosis of pneumonia confirmed by radiological findings documented in their hospital records.

**Exclusion criteria.**

i.  Patients with incomplete or missing relevant medical records.

ii.  Patients diagnosed with COVID-19 or tuberculosis during the study period

iii.  Patients whose pneumonia diagnosis was uncertain or not supported by radiological confirmation.

### Diagnosis of pneumonia

According to the "Guide to antibiotic therapy for common bacterial diseases in Cameroon" [11], pneumonia was clinically diagnosed for patients with fever (39−40°C), chills, tachycardia, cough, dyspnea and chest pain. Hospitalization was considered based on the confusion, respiratory rate, blood pressure and age ≥ 65 years (CRB-65) score. Radiological diagnosis of pneumonia was based on detecting new lung infiltrates/consolidation with patterns and unilateral/bilateral distribution. COVID-19 test was done using Panbio™ SARS-CoV-2 Ag Rapid Test Device (Abbott Rapid Diagnostics, Jena, Germany) for all clinically suspected pneumonia cases during the study period following manufacturer's instructions. Tuberculosis test (microscopy and TB LAMP) was done for patients with typical hallmark signs and symptoms (night sweat, chronic cough lasting >2–3 weeks, gradual weight loss, low-grade fever, hemoptysis etc) strongly suggestive of tuberculosis.

### Ethical consideration

Ethical clearance was obtained from the Institutional Review Board at the Faculty of Health Science, University of Buea (application number 2437−03, approval number 2024/2437-03/UB/SG/IRB/FHS). Administrative authorization was gotten from the regional delegation of public health and director of BRH.

### Data collection and management

A retrospective review of outpatient and inpatient hospital records in BRH was carried out from the 5th of April to the 25th of July 2024 to identify the number of people clinically and radiologically diagnosed with pneumonia during the period of

2020–2023. BRH records from 2020 onward were readily available and accessible for review. In addition, this period overlapped with the COVID-19 pandemic, which had a profound impact on pneumonia diagnosis, hospital turnout, and respiratory disease burden. For inpatient records, all previous files of patients who had been admitted at the private ward, male/female medical wards and VIP unit were reviewed (patients diagnosed with pneumonia are usually hospitalized in these units). In order to ensure confidentiality, the names of the patients on their individual files were masked before handing to the data collector. A competent hospital staff was also assigned at all times to supervise the data collection procedure. Data on the following were collected: patient demographic characteristics, signs and symptoms upon admission, treatment therapy, date of admission, length of hospital stay, date of discharge and state of discharge.

For outpatient records, doctor's consultation registers at the outpatient department were reviewed. Data was collected only for those who were diagnosed with pneumonia and sent home. Data on the following were collected: patient demographic characteristics, signs and symptoms upon admission and treatment therapy.

After reviewing each file and register, all data were entered into Microsoft excel version 2021. Missing or incomplete records were handled by listwise deletion (if any relevant information was lacking in a patient's file, that particular file/patient was dropped out of the study even if other relevant information were available). After entry, data was reviewed for quality by another personnel. All information collected was saved in a designated computer accessible only by authorized personnel concerned with the research.

### Data analysis

Data was analyzed using the Statistical Package for Social Sciences (SPSS) software version 25 (IBM, USA). Sociodemographic characteristics were summarized using frequencies and percentages for categorical variables, and measures of central tendency for continuous variables. Trends in pneumonia cases were represented for various years using line graphs. Pearson's chi-square test was used to compare group proportions. Binary logistic regression analysis was done to investigate associations between the different independent and the dependent (mortality from pneumonia) variables. Only variables with p-value ≤ 0.05 in the bivariate analysis were considered eligible for inclusion in the multivariate regression model. Statistical significance was set at $p < 0.05$.

### Results

A total of 6,998 in-patient hospital files (each file corresponds to a patient) and 11,807 out-patient entries were reviewed for hospitalized and non-hospitalized patients respectively. Out of these, 1,020 were identified as patients diagnosed with pneumonia (Fig 1). After excluding those with incomplete data and those diagnosed with COVID-19 and tuberculosis, 746 files were retained for the study.

The minimum and maximum ages of the retained study population were 20 and 111 years respectively. The mean age was 54.7 ± 19.2 years and the median age was 54.0 (IQR: 39–70) years. There were more females (444 out of 746 [59.5%]) than males. Majority (439 out of 746 [58.8%]) of the pneumonia cases were ≥50 years of age. In terms of occupation, 212 (28.4%) were retired, as shown in Table 1.

As shown in Fig 2, there was a fluctuating trend in pneumonia cases from 2020 to 2023. The highest number of cases were recorded in 2021.

Overall, the number of pneumonia cases fluctuated from January to December in 2020, 2021, 2022 and 2023 (Fig 3). More cases of pneumonia were reported in 2021 especially during the month of September.

After excluding the 84 outpatient pneumonia cases (lack of information with regards to their outcome), the total number of pneumonia cases were brought down to 662. A total of 91 out of 662 admitted pneumonia patients died giving a case fatality rate of 13.7% from 2020–2023. Overall, 55 (60.4%) of those who died were females. The mean age was 64.22 ± 19.4 years and the median age was 70.0 (IQR: 49–81) years. Fig 4 shows the age distribution of admitted patients who died from pneumonia at BRH. It is worth noting that generally the number of people who died increased with age.

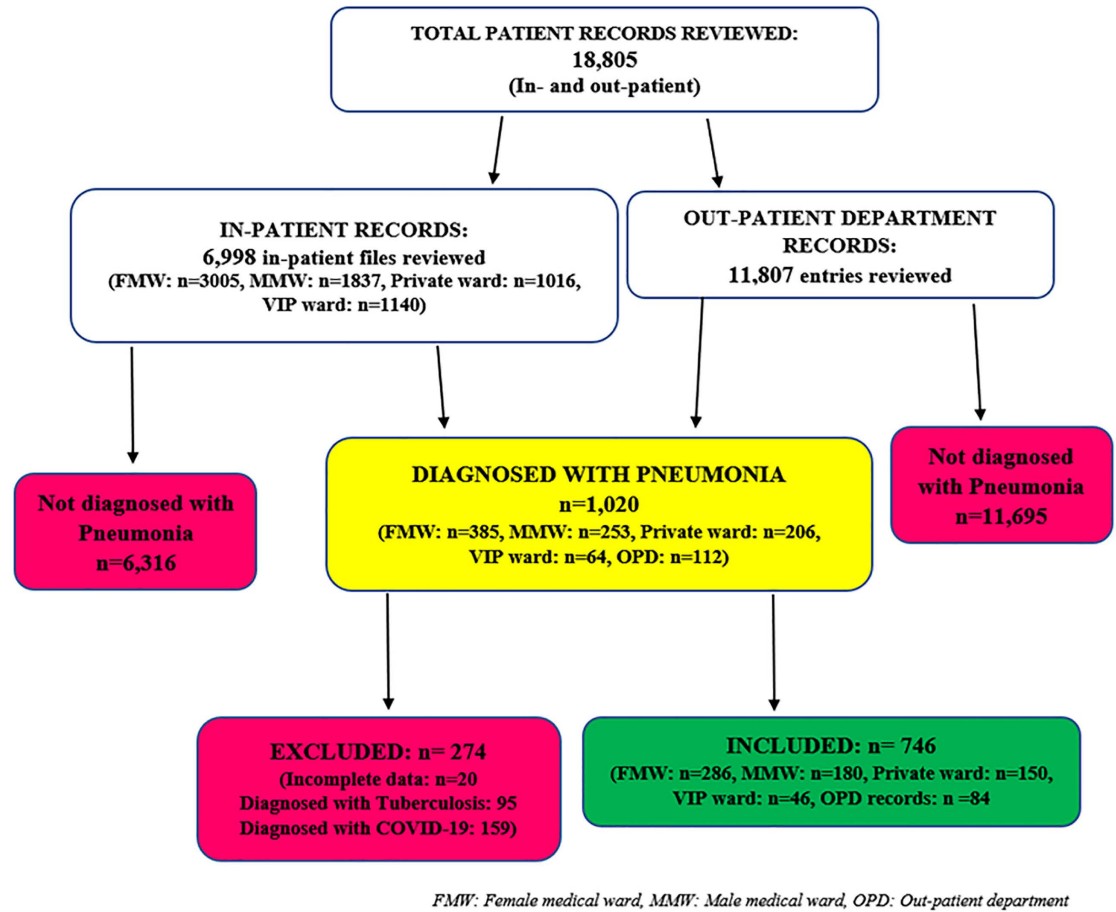

**Fig 1. Review and selection of patient files clinically/radiologically diagnosed with pneumonia at Buea Regional Hospital from 2020 to 2023.**

As shown in Fig 5, there was an overall decreasing trend in the case fatality rate of hospitalized pneumonia patients at BRH from 2020 to 2023 (2021 recorded the highest case fatality rate).

Logistic regression analysis showed that being hospitalized for less than 10 days and not treated with amoxicillin-clavulanic acid were 10 times and 3 times independently and significantly associated with a dead outcome respectively for admitted pneumonia cases (Table 2).

## Discussion

There was a fluctuating trend in pneumonia cases across the years from 2020 to 2023. (and a relatively high mortality rate that seem to decrease between 2020 and 2023). This shows that pneumonia (especially that caused by bacteria) is endemic throughout the year (not a seasonal disease) in BHD corroborating the findings of other studies [12,13]. However, the health system in Cameroon should consider monitoring this disease because there can be a sudden rise in number of cases beyond the expected values due to the presence of some primary infections caused by pathogenic respiratory viruses (some are seasonal) which may weaken the immune system and give rise to typical pneumonia as a secondary infection [14,15]. A good point of reference to corroborate this fact is the rise in number of pneumonia cases in 2021 as seen in this study. Buea health district, just like the rest of Cameroon, recorded its highest number of COVID-19

**Table 1. Sociodemographic characteristics of adults diagnosed with pneumonia at Buea Regional Hospital from 2020 to 2023.**

| Variables (n = 746) | | Number (n) | n (%) |
|---|---|---|---|
| Residence | Others | 264 | 35.3 |
| | Buea town HA | 93 | 12.4 |
| | Bokwango HA | 45 | 6.0 |
| | Buea Road HA | 142 | 19.0 |
| | Tole HA | 11 | 1.4 |
| | Muea HA | 123 | 16.4 |
| | Molyko HA | 68 | 9.1 |
| Gender | Male | 302 | 40.5 |
| | Female | 444 | 59.5 |
| Age Groups (years) | ≥50 | 439 | 58.8 |
| | <50 | 307 | 41.2 |
| Occupation | Business | 102 | 13.6 |
| | Applicant | 26 | 3.4 |
| | Farmer | 108 | 14.4 |
| | House wife | 69 | 9.2 |
| | Retired | 212 | 28.4 |
| | Student | 35 | 4.6 |
| | Teacher | 34 | 4.5 |
| | Others | 160 | 21.4 |
| Marital status | Married | 380 | 50.9 |
| | Single | 216 | 62.4 |
| | Widow | 134 | 17.9 |
| | Widower | 15 | 2.0 |

cases in 2021 [16]. Although we excluded all patients positive for COVID-19 in this study, there is a possibility that a good number of the pneumonia cases diagnosed in 2021 were equally positive for COVID-19 or previously had a recent COVID-19 infection which weakened their immune system [17] and gave way for bacterial pneumonia to set in as a secondary infection.

Variations in many aspects like study population (e.g., children or adults), geographic location, and study period makes it quite challenging to compare our findings with that of other studies. Many studies on pneumonia published online actually address pneumonia in children. In addition, we did not come across any study that investigated the trend in case fatality rate of pneumonia (of bacterial origin) among adults from 2020 to 2023 (this further emphasizes the importance of our study). We however came across studies [18–20] which also revealed a decreasing trend in pneumonia case fatality over time (but not 2020–2023) as seen in our study. This overall decrease in case fatality rate over time could be due to improvements in the diagnosis and management of the disease. There is a possibility that the COVID-19 pandemic led to better triage systems and early recognition of respiratory distress, which also benefited non-COVID pneumonia patients who came to the hospital. The pandemic may have also brought in more emphasis on pulse oximetry and rapid initiation of oxygen in hypoxemic patients. The presence of a pulmonologist at the hospital may also have contributed to better case detection and management, leading to improved survival over time. The general case fatality rate of pneumonia in our study was 13.7%, which is high as compared to that obtained from other studies [18,21,22] conducted in more advanced countries. Majority (72.5%) of the pneumonia patients who died in our study were above 50 years of age and this could be attributed to presence of other co-morbidities thus necessitating prompt diagnosis and early treatment

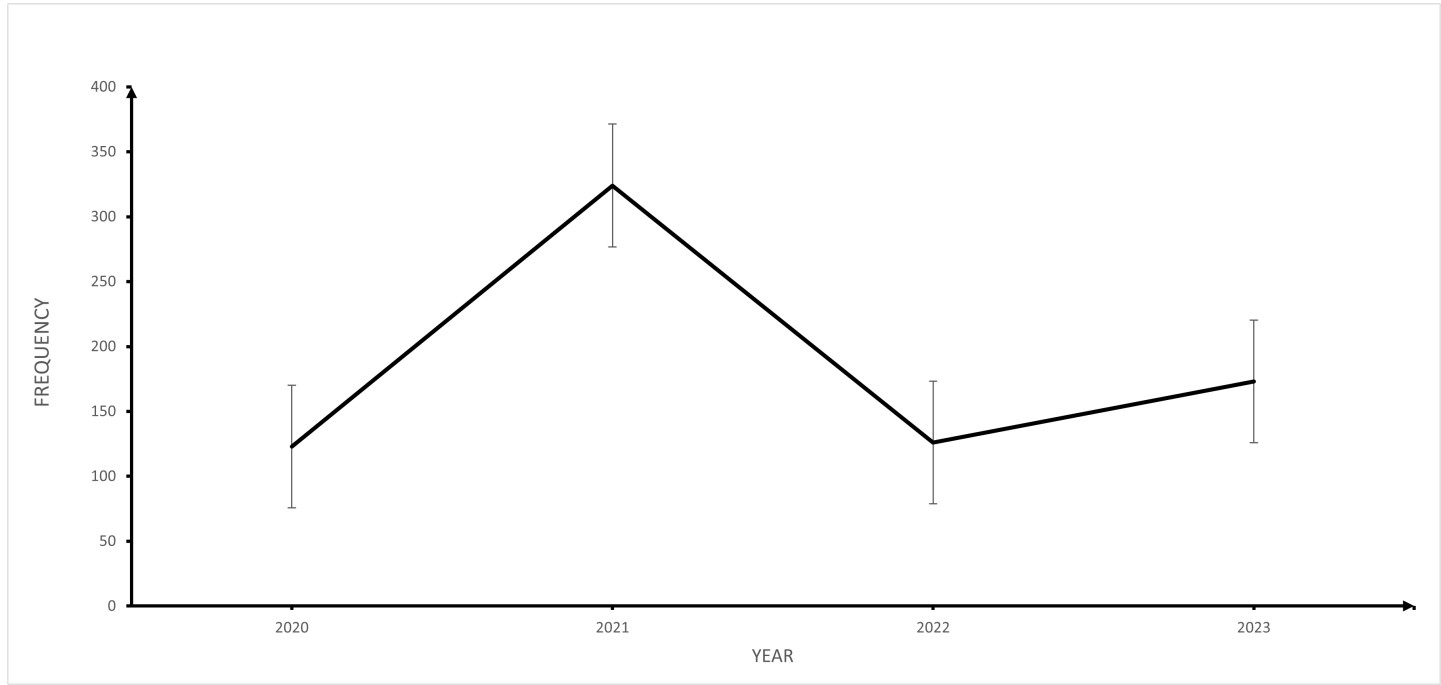

**Fig 2. Trend in pneumonia cases at Buea Regional Hospital from 2020 to 2023.**

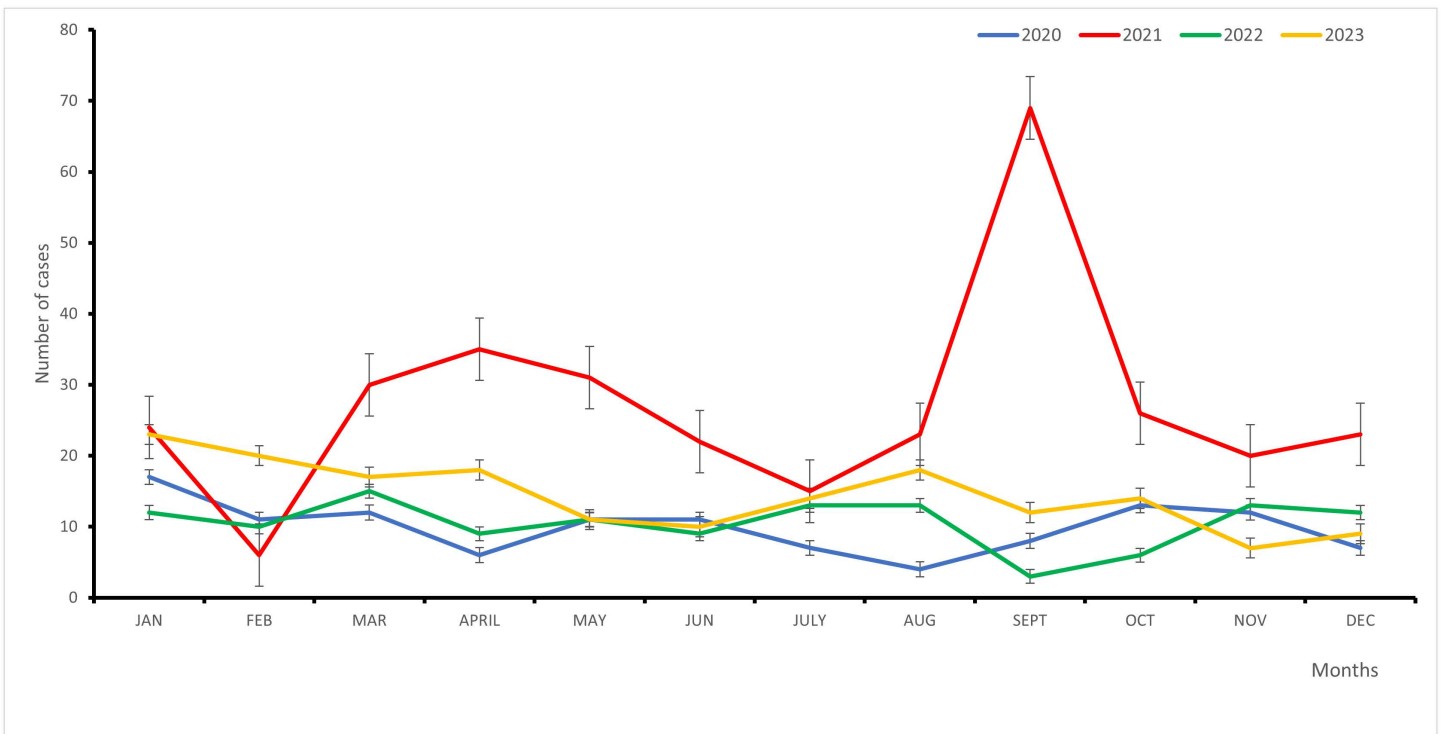

**Fig 3. Trend in pneumonia cases across months of the year in 2020 to 2023 at Buea Regional Hospital.**

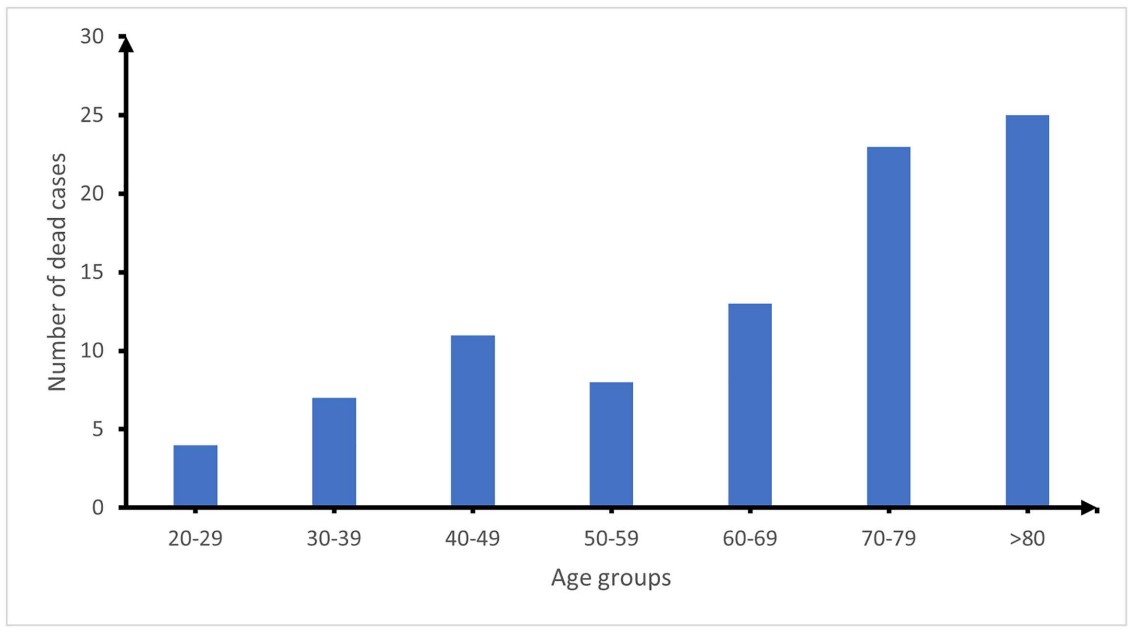

**Fig 4. Age distribution of admitted patients who died from pneumonia at the Buea Regional Hospital from 2020-2023.**

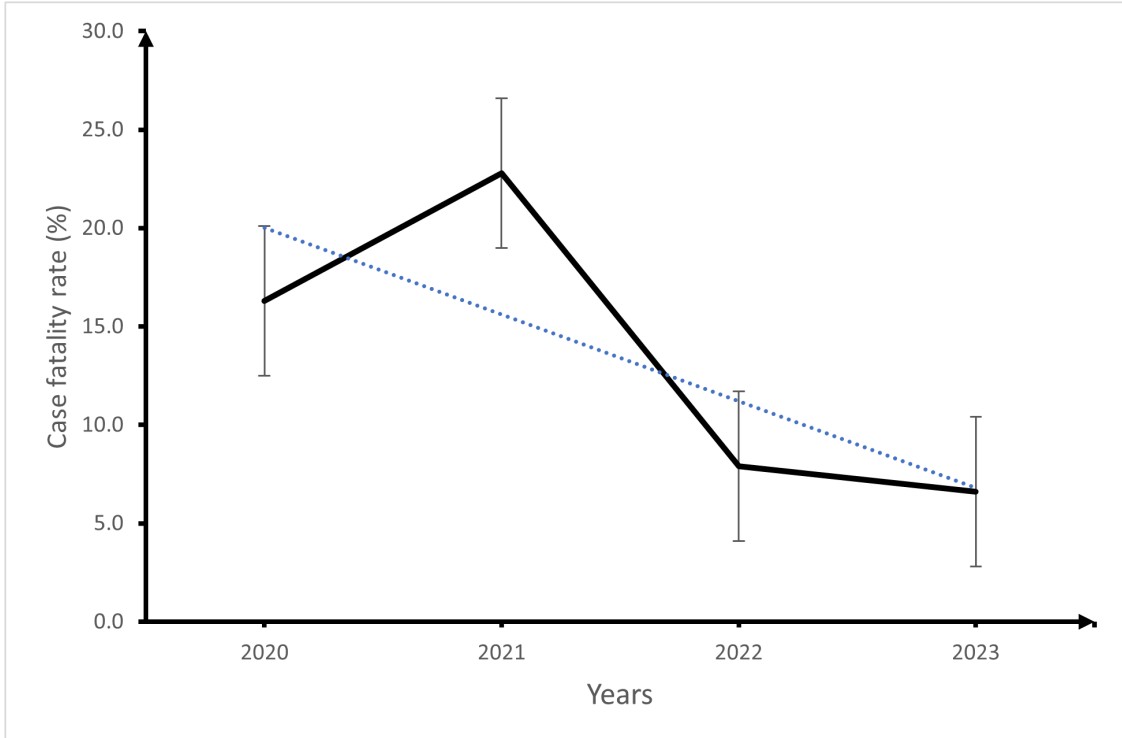

**Fig 5. Trend in the case fatality rates of hospitalized pneumonia patients at the Buea Regional Hospital from 2020-2023.**

**Table 2. Potential factors associated with treatment outcome among patients with pneumonia from 2020-2023 admitted at the Buea Regional Hospital.**

| Variables | Modalities | Total | Treatment outcome (State upon discharge) | | Crude OR (95% CI) | P-value | Adjusted OR (95% CI) | P-value |
|---|---|---|---|---|---|---|---|---|
| | | | Dead | Alive | | | | |
| Gender | Female | 434 | 55 | 379 | 0.77 (0.49-1.21) | 0.269 | | |
| | Male | 228 | 36 | 192 | 1 | | | |
| Age (years) | ≥50 | 428 | 66 | 362 | 1.52 (0.93-2.49) | 0.092 | | |
| | <50 | 234 | 25 | 209 | 1 | | | |
| Duration of hospitalization (Days) | ≤10 | 533 | 88 | 445 | 8.30 (2.58-26.69) | **0.000** | 10.14 (2.31-44.52) | **0.002** |
| | >10 | 129 | 3 | 126 | 1 | | 1 | |
| Comorbidities* | Present | 379 | 55 | 324 | 1.16 (0.74-1.82) | 0.508 | | |
| | Absent | 283 | 36 | 247 | 1 | | | |
| HIV status | Positive | 79 | 13 | 66 | 1.27 (0.67-2.42) | 0.457 | | |
| | Negative | 583 | 78 | 505 | 1 | | | |
| Antibiotic Treatment approach | Monotherapy | 256 | 33 | 223 | 0.90 (0.56-1.41) | 0.612 | | |
| | Combined therapy | 406 | 58 | 348 | 1 | | | |
| Use of Amoxicillin/ clavulanic acid | No | 81 | 20 | 61 | 2.36 (1.34-4.13) | **0.002** | 3.33 (1.43-7.75) | **0.005** |
| | Yes | 581 | 71 | 510 | 1 | | 1 | |

*Includes diabetes, hypertension, chronic kidney disease, cardiovascular disease, chronic lung disease, chronic Liver disease.

OR: Odds ratio CI: Confidence interval.

for a favorable outcome. However, prompt diagnosis and early treatment were most likely not the case for the deceased patients especially those hospitalized for less than 10 days according to our data. This early mortality threshold (≤10 days) aligns with previous findings were 67.3% of in-hospital deaths among pneumonia patients occurred within the first 10 days of admission [23,24]. Studies have shown that most adults in Africa seek medical care only when the disease has become severe and can no more be managed at home [25–28]. Delay in seeking medical care would lead to a delay in diagnosing the disease and eventually a delay in administering the appropriate treatment in time hence leading to the dead of the patient within the first few days of hospitalization.

Our study revealed that not using amoxicillin-clavulanic acid regimen for treatment was independently and significantly associated with mortality from pneumonia. This is similar to the findings of another study [29]. Amoxicillin-clavulanic acid is a recommended empirical treatment for pneumonia [30–32]. Although many other bacteria can cause typical pneumonia, the predominant etiologic agent is usually *Streptococcus pneumoniae* and this bacterium is most often sensitive to amoxicillin [33–35]. The use of stand-alone amoxicillin (without clavulanic acid) is usually appropriate for treating infections caused by *Streptococcus pneumoniae.* However, in order to cover a wider range of bacterial pathogens (e.g., beta-lactamase producing *Staphylococcus aureus, Klebsiella pneumoniae* etc) which could be causing the pneumonia, amoxicillin-clavulanic acid is preferably used [29]. This precaution is usually taken because in many cases, the exact bacterial pathogen causing the pneumonia in question is not habitually known and at times, the pneumonia may be caused by more than one bacterial pathogen (mixed infection). The "Guide to antibiotic therapy for common bacterial diseases in Cameroon" (released in 2024) recommends the use of amoxicillin-clavulanic-acid (as well as other antibiotics) for the treatment of pneumonia. We capitalized on amoxicillin-clavulanic-acid because according to the above-mentioned guide, all the different forms of bacterial pneumonia have amoxicillin-clavulanic-acid as a treatment option (either first choice or second choice). Our study actually showed that if amoxicillin-clavulanic-acid is included in the treatment of pneumonia, there is reduced likelihood of death compared to when it is not included. However, this does not by any chance imply that only amoxicillin-clavulanic-acid should be used for treating pneumonia.

This study had some limitations. Firstly, the retrospective design limits its potential to establish causal relationships. Secondly, the period 2020–2023 overlaps with the COVID-19 pandemic, which may have indirectly affected pneumonia management and hospital resources, thereby influencing outcomes in ways not fully captured in the dataset. Suspected pneumonia cases were all tested for COVID-19 as a routine measure during this period but there is a high probability that some positive COVID-19 cases ended up being part of the study (unintentionally) since the testing method used was the SARS-CoV-2 rapid diagnostic test (RDT) which is less sensitive. Moreover, the testing was done just once even for hospitalized cases who may have ended up contracting the disease during their stay in the hospital. Thirdly, pneumonia diagnosis relied mainly on clinical and radiological findings. Microbiological diagnosis/confirmation of pneumonia is not commonly performed in Cameroon (just like in other Sub-Saharan African countries) due to limited laboratory resources and infrastructure constraints [36,4]. Fourthly, while excluding COVID-19 and TB made the study sample more homogeneous, it also means that the results do not reflect the overall burden of pneumonia at the Buea Regional hospital from 2020 to 2023.

In conclusion, although the number of pneumonia cases fluctuated over time, they remained consistently present across all months from 2020 to 2023, highlighting the year-round nature of pneumonia transmission and its endemicity in the Buea Health District. The general case fatality rate for hospitalized pneumonia patients was 13.7% with an overall decreasing trend from 2020 to 2023. Being hospitalized for less than 10 days and not treating with amoxicillin-clavulanic acid were both independently and significantly associated with mortality from pneumonia at BRH. These findings underscore the importance of timely diagnosis and empiric treatment with guideline recommended agents such as amoxicillin–clavulanic acid so as to reduce pneumonia mortality at the Buea regional hospital.

## Supporting information

**S1 File. Raw spreadsheet data of pneumonia cases at Buea Regional hospital from 2020 to 2023.**
(XLSX)

**S2 File. Trends in pneumonia cases and mortality from 2020 to 2023 at Buea Regional Hospital.**
(XLSX)

## Acknowledgments

We acknowledge the nurses and the data manager of the Buea Regional Hospital for their support in retrieving past data. Special thanks also go to the administrative staff of the hospital for giving us permission to have access to their data

## Author contributions

**Conceptualization:** Kukwah Anthony Tufon, Tiayah Patience Foumene.

**Data curation:** Kukwah Anthony Tufon, Tiayah Patience Foumene.

**Formal analysis:** Kukwah Anthony Tufon, Malika Esembeson.

**Investigation:** Tiayah Patience Foumene, Tiayah Precious Yemene, Chapajong Mariecole Nguatem, Ronald Mbua Gobina, Nkouonlack Cyrille, Tiayah Munge.

**Methodology:** Tiayah Patience Foumene, Malika Esembeson, Ayah Flora Bolimo, Tiayah Precious Yemene, Teuwafeu Denis Georges.

**Resources:** Ngomba Divine Martin Mokake.

**Supervision:** Kukwah Anthony Tufon, Chapajong Mariecole Nguatem, Ngomba Divine Martin Mokake.

**Validation:** Kukwah Anthony Tufon, Malika Esembeson, Ayah Flora Bolimo, Ngomba Divine Martin Mokake.

**Visualization:** Kukwah Anthony Tufon.

**Writing – original draft:** Kukwah Anthony Tufon, Tiayah Patience Foumene.

**Writing – review & editing:** Kukwah Anthony Tufon.

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
