## [Decision Letter · Decision Letter 0]

5 Aug 2025

Dear Dr.  Tufon,

We look forward to receiving your revised manuscript.

Kind regards,

Leonard Ighodalo Uzairue, PhD

Academic Editor

PLOS ONE

Additional Editor Comments (if provided):

1. Thank you for submissing your manuscript to PLOSONE. There are some merit in the work, however, there are significant issue in the write-up. There is need so check grammar across the manuscript and also make sure to italize where necessary.

2. There is need to expand on the inclusion and exclusion for the study

Reviewers' comments:

Reviewer's Responses to Questions

**Comments to the Author**

1. Is the manuscript technically sound, and do the data support the conclusions?

Reviewer #1: Partly

Reviewer #2: Partly

Reviewer #3: Yes

2. Has the statistical analysis been performed appropriately and rigorously?

Reviewer #1: Yes

Reviewer #2: No

Reviewer #3: I Don't Know

3. Have the authors made all data underlying the findings in their manuscript fully available?

Reviewer #1: No

Reviewer #2: Yes

Reviewer #3: Yes

4. Is the manuscript presented in an intelligible fashion and written in standard English?

Reviewer #1: Yes

Reviewer #2: No

Reviewer #3: Yes

Reviewer #1: General comments:

Distinguishing between typical and atypical pneumonia clinically is difficult; for example, patient may present with dry cough and low-grade fevers (criteria used to define atypical pneumonia) and still have findings on chest x-ray that are characteristic of typical pneumonia. Patients with COVID-19 diagnosis were excluded from the analysis but it is unclear if COVID testing was offered to all patients with suspected pneumonia. It is highly plausible that a significant number of patients with typical/atypical pneumonia diagnosis had COVID. I would recommend combining all the pneumonia cases and redoing analysis.

Need to discuss limitations of the study.

See the attachment for the rest of the comments.

Reviewer #2: This is a well-executed, relevant, and timely retrospective study assessing the trends and predictors of pneumonia mortality in a low-resource setting. It addresses a crucial public health concern and fills a significant gap in the regional literature, particularly from Sub-Saharan Africa. The methodology is robust, and the findings are valuable for healthcare planning and clinical practice. However, there is some room for improvement.

Here are my comments:

1. Introduction

Rephrase verbose or redundant parts, e.g., "Pneumonia cases like wise deaths..." to "Pneumonia cases and associated deaths..."

Use more precise language: "It proves to us..." to "It suggests..." or "It indicates..."

Avoid generalizations without sources, e.g., “...helps in predicting future outbreaks...” (requires clarification or reference).

2. Methods

Clarify diagnostic criteria for classifying typical vs. atypical pneumonia—were these physician-noted, ICD-coded, or based on lab/radiology?

It’s unclear how pneumonia types were classified in outpatient cases without radiological confirmation.

Mention how missing or incomplete records were handled statistically (listwise deletion or imputation?).

3. Results

Provide numbers alongside percentages throughout. For instance, instead of only “13.7% fatality rate,” also state “91 out of 662.”

If possible, distinguish between univariate and multivariate analysis outcomes.

4. Discussion

Avoid speculative language like "proves to us". Replace with "suggests" or "indicates."

Clarify limitations explicitly, e.g.:

A.Retrospective design limits causality.

B. Lack of pathogen confirmation (no lab cultures).

C. Possible misclassification bias due to reliance on clinical diagnoses.

Consider adding a brief note on potential interventions (e.g., increased community awareness, pneumococcal vaccination, antibiotic stewardship).

5. Conclusion

Consider including a final sentence emphasizing policy or clinical recommendations for Buea Health District or similar contexts.

Reviewer #3: Dear author(s),

Congratulations on your study examining mortality trends and associated factors in patients with typical and atypical pneumonia in Cameroon. I have read your study carefully. Please find my suggestions and questions below.

Is there a specific reason for examining harvests from 2020 to 2023?

Line 62: ‘Generally, pneumonia is usually classified as typical or atypical’ should be changed to ‘Pneumonia is classified as typical or atypical based on its clinical features.’

You provided the rationale in the last paragraph of the introduction. You should write the purpose of your study at the end.

In the materials and methods section, you should describe the clinical characteristics of typical and atypical pneumonia (provide definitions). When providing these definitions, it would be helpful to mention the guidelines used by doctors in your country for the diagnosis and treatment of pneumonia.

You found that mortality increases in patients with a hospital stay of less than 10 days. Why did you set the hospital stay at 10 days? Do you have any references for this?

The first paragraph of the Discussion section should summarise the major results of your study. Then, you should discuss each finding in a separate paragraph.

Sincerely.

**Do you want your identity to be public for this peer review?** For information about this choice, including consent withdrawal, please see our Privacy Policy

Reviewer #1: No

Reviewer #2: No

Reviewer #3: **Yes: ** Ayse Baha

---

## [Author Response · Author response to Decision Letter 1]

21 Sep 2025

RESPONSE TO REVIEWERS

RESPONSE

Manuscript has been revised to meet PLOS ONE's style requirements

Additional Editor Comments (if provided):

1. Thank you for submissing your manuscript to PLOSONE. There are some merit in the work, however, there are significant issue in the write-up. There is need so check grammar across the manuscript and also make sure to italize where necessary.

RESPONSE

We have checked and corrected grammatic errors to the best of our knowledge

2. There is need to expand on the inclusion and exclusion for the study

RESPONSE:

More detailed inclusion and exclusion criteria not stated in the methods (line 101-110)

REVIEWER #1:

General comments:

Distinguishing between typical and atypical pneumonia clinically is difficult; for example, patient may present with dry cough and low-grade fevers (criteria used to define atypical pneumonia) and still have findings on chest x-ray that are characteristic of typical pneumonia. Patients with COVID-19 diagnosis were excluded from the analysis but it is unclear if COVID testing was offered to all patients with suspected pneumonia. It is highly plausible that a significant number of patients with typical/atypical pneumonia diagnosis had COVID. I would recommend combining all the pneumonia cases and redoing analysis.

Need to discuss limitations of the study.

RESPONSE:

Thank you for your valuable suggestion. We have now grouped atypical and typical pneumonia. Its just referred to as “pneumonia” at this point.

We actually highlighted the limitations of the study in the different paragraphs (where applicable) at the discussion section. May be that wasn’t clear enough. We have now dedicated a paragraph for limitations of the study in the discussion section (line 300-316)

Specific comments:

43…………………………………………………………………..This implies there

44 is need for prompt diagnosis and treatment of Pneumonia cases with amoxicillin

clavulanic acid in order to reduce the case fatality rate in Buea Health District

There are many appropriate antibiotics for treatment of pneumonia and would not limit the recommendation to amoxicillin-clavulanic.

RESPONSE:

Thank you for your remark. The “Guide to antibiotic therapy for common bacterial diseases in Cameroon” (released in 2024) recommends the use of amoxicillin-clavulanic-acid (as well as other antibiotics) for the treatment of pneumonia. We capitalized on amoxicillin-clavulanic-acid because according to the above-mentioned guide, all the different forms of bacterial pneumonia have amoxicillin-clavulanic-acid as a treatment option (either first choice or second choice). Our study actually shows that if amoxicillin-clavulanic-acid is included in the treatment, there is a higher chance of avoiding dead as a result of the pneumonia. In summary, the recommendation we made in our study is strictly based on our findings and IT DOES NOT BY ANY CHANCE imply that only amoxicillin-clavulanic-acid should be used for treating pneumonia. We have however rephrased the sentence (line 42 to 45)

97………………………………………………………………………………………..Only patients clinically

98 and radiologically diagnosed with typical or atypical pneumonia (excluding tuberculosis)

99 were considered for this study.

Can you provide details on criteria used for clinical diagnosis of pneumonia, atypical pneumonia and how pneumonia was differentiated from TB, COVID or other viral pneumonia using clinical diagnosis. Also, it is not clear from the results which patients were diagnosed using clinical criteria vs. radiographic criteria vs. both.

RESPONSE

Thank you for your comment. We have now included in the methods a subheading describing how diagnosis pneumonia, COVID-19 and Tuberculosis were considered (line 111-123)

142…………………Out of these, 1,020 were identified as patients diagnosed with Pneumonia

143 (fig 1). After excluding those with incomplete data and those diagnosed with COVID-19

and Tuberculosis, 746 files were retained for the study.

I think it would be helpful if you can talk about some challenges using clinical diagnosis. For example, someone diagnosed with pneumonia may end up being diagnosed with TB and differentiating between typical and atypical pneumonia may not be possible using clinical diagnosis alone. Also, in figure 1, 159 patients diagnosed with pneumonia were excluded because they had COVID-19. Was COVID-19 testing offered to all patients with pneumonia? If not, this is a limitation that needs to be addressed as other patients with COVID pneumonia are most likely represented in the 746 participants included in the analysis.

RESPONSE:

Thank you for pointing this out. This has been included as a limitation to the study in the designated paragraph under discussion section. (Line 309-316)

Figure one: blurry, please submit a clearer image.

RESPONSE

A clearer image has now been submitted

170……………………………………………More cases of typical pneumonia were reported in

2021 especially during the month of September.

How do you explain this? Was there a change in how the hospital defined pneumonia? Could this be due to COVID-19?

RESPONSE:

This was most likely due to COVID-19. Check the first paragraph of the discussion which also stands as a limitation of the study

216 Logistic regression analysis showed that being hospitalized for less than 10 days and not treated with amoxicillin-clavulanic acid were 10 times and 3 times independently and

significantly associated with a dead outcome respectively for admitted pneumonia

It seems like the majority 88% (581/662) of participants were treated with antibiotics (256 monotherapy + 406 combined) What other antibiotics were prescribed besides amoxicillin-clavulanic?

RESPONSE

Other antibiotics like Azithromycin, metronidazole, amoxicillin etc were equally prescribed. However, we focused on amoxicillin-clavulanic-acid because according to The “Guide to antibiotic therapy for common bacterial diseases in Cameroon”, all the different forms of bacterial pneumonia have amoxicillin-clavulanic-acid as a treatment option (either first choice or second choice) so we wanted to really see how useful or relevant amoxicillin-clavulanic-acid is in this context.

241…. Although we excluded all patients positive for COVID-19 in this study, there is a possibility that a good number of the typical/atypical pneumonia cases diagnosed in 2021 were equally positive for COVID-19 or previously had a recent COVID-19 infection which weakened their immune system and gave way for bacterial pneumonia to set in as a secondary infection.

COVID-19 in itself can cause pneumonia, please revise. Also, was the policy in BHD to test all patients admitted with pneumonia for COVID-19? If not, it would seem like other patients with COVID-19 pneumonia were included in the analysis.

RESPONSE

We know that COVID-19 causes pneumonia. However, it was not part of our interest. During the said period, all clinically suspected pneumonia cases were tested for COVID-19. Unfortunately, we can’t say with certainty that we successfully excluded all COVID-19 cases from the study so this just has to come up as one of the limitations of the study. Check the paragraph addressing limitations in the discussion section.

252 …………………………We however came across studies [16–18] which also revealed a decreasing trend in pneumonia case fatality over time (but not 2020 to 2023) as seen in our study. This overall decrease in case fatality rate over time could be due to improvements in the diagnosis and management of the disease.

There’s a notable decrease in trends in pneumonia case fatality even though the total number of patients with pneumonia remained the same. Are you able to provide additional details on what the hospital was/is doing to improve diagnosis and management?

RESPONSE

Thank you for pointing this out. There is a possibility that the COVID-19 pandemic led to better triage systems and early recognition of respiratory distress, which also benefited non-COVID pneumonia patients who came to the hospital. The pandemic may have also brough in more emphasis on pulse oximetry and rapid initiation of oxygen in hypoxemic patients. The presence of a pulmonologist at the hospital may also have contributed to better case detection and management, leading to improved survival over time.

This has been added in the discussion section of the manuscript (line 258-263)

258 Majority (72.5%) of the pneumonia patients who died in our study were above 50 years of age and this could be attributed to weakened immune systems thus necessitating prompt diagnosis and early treatment for a favorable outcome.

Do you have any data on the types and frequency of pathogens isolated from patients with pneumonia? For those pathogens with available vaccines (e.g., pneumococcal pneumonia, H. flu, COVID, influenza), vaccination may offer better protection and overall reduction in sickness and death.

RESPONSE

Thank you for pointing this out. This information is lacking. Check the paragraph addressing limitations in the discussion section.

275…………………In fact, combining clavulanic acid to amoxicillin does not come with any added advantage (neither does it come with a disadvantage) for the elimination of Streptococcus pneumoniae[33].

This is not accurate. Some patients could have been infected with resistant bacteria that will respond to amoxicillin-clavulanic over amoxicillin alone. Unless you have drug sensitivity data on all pneumonia isolate, I would not say that Amox-clav does not add any advantage. Please revise.

RESPONSE

The statement you mentioned above is accurate as long as the organism in question is Streptococcus pneumoniae. We talked about this lack of advantage of clavulanic acid addition strictly for Streptococcus pneumoniae. Amoxicillin Resistance in S. pneumoniae is not due to β-lactamase production. Instead, resistance arises from altered PBPs (target modification). Since clavulanic acid only blocks β-lactamase enzymes and S. pneumoniae does not produce them, adding clavulanic acid does not enhance amoxicillin’s activity against S. pneumoniae. However, as clearly written in the manuscript, there is added advantage when we start considering other bacterial pathogens that may cause pneumonia that is why the use of Amox-clav combination is still a better approach. Please read the paragraph in question till the end. However, since you seem not to be comfortable with that particular sentence, the sentence has been deleted in the revised manuscript. Thank you

REVIEWER #2:

This is a well-executed, relevant, and timely retrospective study assessing the trends and predictors of pneumonia mortality in a low-resource setting. It addresses a crucial public health concern and fills a significant gap in the regional literature, particularly from Sub-Saharan Africa. The methodology is robust, and the findings are valuable for healthcare planning and clinical practice. However, there is some room for improvement.

Here are my comments:

1. Introduction

Rephrase verbose or redundant parts, e.g., "Pneumonia cases like wise deaths..." to "Pneumonia cases and associated deaths..."

Use more precise language: "It proves to us..." to "It suggests..." or "It indicates..."

Avoid generalizations without sources, e.g., “...helps in predicting future outbreaks...” (requires clarification or reference).

RESPONSE

Thank you for pointing these out all the corrections have been made. As for the “...helps in predicting future outbreaks...”, that statement is just an obvious fact in epidemiology and can apply for any disease as long as it shows a periodic pattern in its occurrence over time. If we see a pattern in the occurrence of a disease over the years e.g the number of cases always rise above expected in the month of March, we can comfortably predict that the disease would have an epidemic or an outbreak in the upcoming month of March 2026. We might be wrong but we don’t think that needs a reference. It’s an obvious logic.

2. Methods

Clarify diagnostic criteria for classifying typical vs. atypical pneumonia—were these physician-noted, ICD-coded, or based on lab/radiology?

RESPONSE

Thank you for pointing this out. A subheading (Diagnosis of pneumonia) has been added in the methods section to address this (line 111-123)

It’s unclear how pneumonia types were classified in outpatient cases without radiological confirmation.

RESPONSE

Diagnosis of pneumonia in this study was considered only when both clinical and radiological findings were available to confirm it. Every out-patient case without radiological confirmation records were considered as patient with missing/incomplete data and so was not included in the 746 cases (listwise deletion)

Mention how missing or incomplete records were handled statistically (listwise deletion or imputation?).

RESPONSE

Thank you for pointing this out. We handled missing or incomplete records by listwise deletion (line 149-151)

3. Results

Provide numbers alongside percentages throughout. For instance, instead of only “13.7% fatality rate,” also state “91 out of 662.”

RESPONSE

Numbers have now been provided alongside percentages were applicable

If possible, distinguish between univariate and multivariate analysis outcomes.

RESPONSE

All the relevant details is seen in table 2 (crude odds ratio is the univariate analysis while adjusted odds ratio is the multivariate analysis. Only variables with p-value ≤ 0.05 in the bivariate analysis were considered eligible for inclusion in the multivariate regression model

4. Discussion

Avoid speculative language like "proves to us". Replace with "suggests" or "indicates."

RESPONSE

Speculative languages have been removed

Clarify limitations explicitly, e.g.:

A.Retrospective design limits causality.

B. Lack of pathogen confirmation (no lab cultures).

C. Possible misclassification bias due to reliance on clinical diagnoses.

Consider adding a brief note on potential interventions (e.g., increased community awareness, pneumococcal vaccination, antibiotic stewardship).

RESPONSE

An entire paragraph in the discussion section have been dedicated for limitations (line 300-316)

5. Conclusion

Consider including a final sentence emphasizing policy or clinical recommendations for Buea Health District or similar contexts.

RESPONSE

To the best of our knowledge, the last sentence in the conclusion paragraph addresses this need.

REVIEWER #3:

Dear author(s),

Congratulations on your study examining mortality trends and associated factors in patients with typical and atypical pneumonia in Cameroon. I have read your study carefully. Please find my suggestions and questions below.

Is there a specific reason for examining harvests from 2020 to 2023?

RESPONSE

Thank you for this question. Hospital records from 2020 onward were readily available, and accessible for review. In addition, this period overlaps with the COVID-19 pandemic, which had a profound impact on pneumonia diagnosis, hospital attendance, and respiratory disease burden. A 4-year span provides a reasonable time frame to observe trends, fluctuations, and seasonal variations, while keeping the dataset manageable and relevant

Line 62: ‘Generally, pneumonia is usually classified as typical or atypical’ should be changed to ‘Pneumonia is classified as typical or atypical based on its clinical features.’

RESPONSE

Statement has been changed

You provided the rationale in the last paragraph of the introduction. You should write the purpose of your study at the end.

RESPONSE

The last sentence in the introduction section now carries the purpose/aim of the research (line 85-87)

In the materials and methods section, you should describe the clinical characteristics of typical and atypical pneumonia (provide definitions). When providing these definitions, it would be helpful to mention the guidelines used by doctors in your country

---

## [Decision Letter · Decision Letter 1]

14 Oct 2025

Dear Dr. Tufon,

We look forward to receiving your revised manuscript.

Kind regards,

Leonard Ighodalo Uzairue, PhD

Academic Editor

PLOS ONE

Journal Requirements:

Reviewers' comments:

Reviewer's Responses to Questions

**Comments to the Author**

Reviewer #1: (No Response)

Reviewer #2: All comments have been addressed

2. Is the manuscript technically sound, and do the data support the conclusions?

Reviewer #1: (No Response)

Reviewer #2: (No Response)

3. Has the statistical analysis been performed appropriately and rigorously?

Reviewer #1: (No Response)

Reviewer #2: (No Response)

4. Have the authors made all data underlying the findings in their manuscript fully available?

Reviewer #1: (No Response)

Reviewer #2: (No Response)

5. Is the manuscript presented in an intelligible fashion and written in standard English?

Reviewer #1: (No Response)

Reviewer #2: (No Response)

Reviewer #1: This is a much better version of the manuscript than the first one, but additional revisions are still needed. Below are some recommendations that can strengthen the paper. Consider a review by someone with experience in technical editing before resubmission. See the attachment for more detailed comments.

Reviewer #2: (No Response)

**Do you want your identity to be public for this peer review?** For information about this choice, including consent withdrawal, please see our Privacy Policy

Reviewer #1: No

Reviewer #2: No

---

## [Author Response · Author response to Decision Letter 2]

17 Oct 2025

RESPONSE TO REVIWERS

JOURNAL REQUIREMENTS:

RESPONSE:

Recommendation to cite specific previously published works was reviewed and deemed relevant

RESPONSE:

We have reviewed our reference list and to he best of our knowledge, none of the references have been retracted. However, if you think otherwise, please indicate to us the particular retracted reference so that we can do the needful.

REVIEWER

This is a much better version of the manuscript than the first one, but additional revisions are still needed. Below are some recommendations that can strengthen the paper. Consider a review by someone with experience in technical editing before resubmission.

RESPONSE

Thank you for your dedication in reading and correcting this work.

29 Only patients clinically and radiologically diagnosed with pneumonia of bacterial origin (excluding tuberculosis) were considered for this study.

Consider revising to “Only patients clinically and radiologically diagnosed with pneumonia (excluding TB and COID-19) were considered for this study”. Other viruses can cause pneumonia and since you only tested for TB and COVID, it is best to keep this broad and not limit to pneumonia with bacterial origin.

RESPONSE

Correction done as requested. (line 29 -30)

35 …. There was a decreasing trend in case fatality rate across the years with a general case fatality rate of 13.7%.

Need to add the case fatality rate by year to show the trend.

RESPONSE

Correction done as requested. (line 35 -36)

39… There was a fluctuating trend in pneumonia cases in Buea regional hospital from 2020 to 2023 implying that pneumonia is endemic in Buea Health district

I believe the intend is to highlight the year-round (as opposed to seasonal) high rates of pneumonia infection in the district. The above sentence does not bring this out, please revise

RESPONSE

Correction done as requested. (line 40 -42)

61………………… …………………………………..It can be caused by several bacteria, viruses and fungi [7]. Generally, pneumonia is classified as typical or atypical based on its clinical features or causative agent. Typical pneumonia refers to a more severe form of pneumonia usually caused by Streptococcus pneumoniae (majority of the cases), Staphylococcus aureus, Haemophilus influenzae and other Gram-negative bacteria that can survive in the presence of oxygen [8].

I would recommend removing the distinction between typical and atypical pneumonia and just focusing on pneumonia as diagnosed by clinical and radiographic criteria. You’ve already done so in the result section (for the most part). The distinction does not help in addressing the main objectives of the paper (identifying the trend of pneumonia and factors associated with pneumonia mortality at the Buea regional hospital from 2020 to 2023). You can still include the leading causes of pneumonia that you’ve highlighted above.

RESPONSE

Correction done as requested. (line 64 -66)

160.. using line graphs, while description of the various types of pneumonia across the years

As stated above, the distinction between typical and atypical pneumonia does not add clarity but brings with it confusion given lack of empirical evidence that can allow you to make the distinction.

RESPONSE

Correction done as requested. (line 156 -157)

173 Figure 1: Review and selection of patient files clinically/radiologically diagnosed with…

Figure 1 is not included. Again, would recommend changing the title to “…patient files clinically/radiologically diagnosed with typical and atypical pneumonia..” similar to the title of Table 1.

RESPONSE

Correction done as requested. (line 168 -169)

235 This shows that pneumonia (especially that caused by bacteria) is endemic throughout the year (not a seasonal disease) in BHD corroborating the findings of other studies

You should also include right in the beginning of discussion a summary of the other key findings of the study (relatively high mortality rate that seem to decrease between 2020 and 2023).

RESPONSE

Correction done as requested. (line 234 -235).

246 …were equally positive for COVID-19 or previously had a recent COVID-19 infection which weakened their immune system and gave way for bacterial pneumonia to set in as a secondary infection…

Please add a reference about recent COVID infection weakening patients immune system.

RESPONSE

Reference has added as requested. (line 248)

265 ……. Majority (72.5%) of the pneumonia patients who died in our study were above 50 years of age and this could be attributed to weakened immune systems.

Recommend changing “weakened immune systems” to presence of other co-morbidities. This is broad and not as specific as weekend immune system.

RESPONSE

Correction done as requested. (line 268)

284… stand-alone amoxicillin (without clavulanic acid) is usually good enough for eliminating Streptococcus pneumoniae

Recommend changing “good enough for eliminating” to something like appropriate for treating Streptococcus… Elimination can mean different things clinically and microbiologically.

RESPONSE

Correction done as requested. (line 285 -286)

297….. there is a higher chance of avoiding dead as a result of the pneumonia

Consider revising to something like there is reduced likelihood of death compared to when this antibiotic is not used.

RESPONSE

Correction done as requested. (line 297 -299)

312……………… As such, its quite difficult to distinguish between typical and atypical pneumonia

No need to include this, please delete.

RESPONSE

Correction done as requested.

---

## [Editor Report · Decision Letter 2]

29 Oct 2025

Trends and factors associated with Pneumonia mortality at Buea regional hospital in Cameroon from 2020 to 2023

PONE-D-24-47412R2

Dear Dr. Tufon,

We’re pleased to inform you that your manuscript has been judged scientifically suitable for publication and will be formally accepted for publication once it meets all outstanding technical requirements.

Kind regards,

Leonard Ighodalo Uzairue, PhD

Academic Editor

PLOS ONE
---

## [Editor Report · Acceptance letter]

PONE-D-24-47412R2

PLOS ONE

Dear Dr. Tufon,

I'm pleased to inform you that your manuscript has been deemed suitable for publication in PLOS ONE. Congratulations! Your manuscript is now being handed over to our production team.

Kind regards,

on behalf of

Dr. Leonard Ighodalo Uzairue

Academic Editor

PLOS ONE